



# Hidden Hazards: the conditions that potentially enabled the mudflow disaster at Villa Santa Lucía in Chilean Patagonia

Marcelo A. Somos-Valenzuela[1,2] , Joaquín E. Oyarzún-Ulloa[3], Ivo J. Fustos-Toribio[3*] , Natalia Garrido-Urzua[4], Chen Ningsheng[5,6]

[1] Department of Forest Sciences, Faculty of Agriculture and Forest Sciences, Universidad de La Frontera, Av. Francisco Salazar 01145, Temuco, Chile, 4780000

[2] Butamallin Research Center for Global Change, Universidad de La Frontera, Av. Francisco Salazar 01145,
Temuco, Chile, 4780000

[3] Department of Civil Engineering, Universidad de La Frontera, Av. Francisco Salazar 01145, Temuco, Chile, 4780000

[4] Servicio Nacional de Geologia y Mineria, Av. La Paz 406, Puerto Varas, Región de los Lagos, Chile, 5550000

[5] Institute of Mountain Hazards and Environment, Chinese Academy of Sciences, Chengdu 610041, China;
[6] University of Chinese Academy of Sciences, Beijing 100049, China;

*Corresponding author:* Ivo J. Fustos-Toribio (ivo.fustos@ufrontera.cl)



**Abstract.** The evaluation of potential mass wasting in mountain areas is a very complex process because there is not enough information to quantify the probability and magnitude of these events. Identifying the whole chain of events

is not a straightforward task, and the impacts of mass wasting processes depend on the conditions downstream of the origin. Additionally, climate change is playing an essential role in the occurrence and distribution. Mean temperatures are continuously rising to produce long term instabilities, particularly on steep slopes. Extreme precipitations events are more recurrent as well as heat waves that can melt snow and glaciers, increasing the water available to unstabilized slopes.

In this paper, we present an example that portraits the complexities in the evaluation of the chain of events. On the 16 of December of 2017, a rockslide occurred in the Yelcho mountain range. In that event, 7 million m$^3$ of rocks and soil fell on the Yelcho glacier depositing 2 million m$^3$ on the glacier terminal, and the rest continued downstream, triggering a mudflow that hit Villa Santa Lucia in the Chilean Patagonia, killing 22 people. The rockslide event or similar was anticipated in the region by the National Geological and Mining Survey (Sernageomin in Spanish).

However, the effects of the terrain characteristics along the runout area were more significant than what was anticipated. In this work, we evaluate the conditions that enable the mudflow that hits Villa Santa Lucia. We used the information generated by Sernageomin's professional after the mudflow. We carried out geotechnical tests to characterize the soil. We simulated the mudflow using two hydrodynamics software (r-avaflow and Flo-2D) that can handle the rheology of the water-soil mixture.

Our results indicate that the soil is classified as volcanic pumices. This type of soil can be susceptible to the collapse of the structure when subjected to shearing (molding), flowing like a viscous liquid. From the numerical modeling, we concluded that r-avaflow performs better than Flo2D. We can reproduce the mudflow satisfactorily using water content in the mixture ranging from 30 to 40%. Finally, in order to achieve the water content, we need a source of water smaller than 3 million m$^3$ approximately. From the simulations and soil tests, we determined that in the area

scoured by the mudflow, there were around 2,789,500 m$^3$ of water within the soil. Therefore, the conditions of the valley were crucial to enhance the impacts of the landslide. This result is relevant because it highlights the importance of evaluating the complete chain of events to map hazards. We suggest that in future hazard mapping, geotechnical studies in combination with hydrodynamic simulation should be included, in particular, when human lives are at risk.

Keywords: Geological hazards, r.avaflow, mudflow modeling, Southern Andes mudflows


## 1 Introduction

Climate change has a causal link to landslides in mountain environments due to the increase of extreme precipitation mean, minimum-maximum temperature, and heatwaves (Drewes et al., 2018; Huggel et al., 2012; Stoffel and Huggel, 2012; Tichavský et al., 2019). Because this is of great concern for high slope instabilities, we need to understand the concepts of long-term disposition such as geology (lithology, layering, faults), topography (vertical extent, slope) and ice conditions (ice cover, hanging glaciers, permafrost), short-term disposition that can change long term disposition such as climate change, and trigger effects including earthquakes, extreme snowmelt, heat waves, and heavy precipitation that lead to discrete events (Fischer et al., 2012, 2013; Haeberli, 2013; Huggel et al., 2010; Tichavský et al., 2019). In our contribution, we will evaluate the generation of cascade events associated with the Villa Santa Lucia mudflow in the Northern Patagonia. Our analysis will be carried out through the integration of geotechnical and in-situ data with numerical modelling to evaluate the conditions that can generate this events in the Southern Andes.

Temperature affects rock/ice slope at different time scales. In the short-term, temperatures affect slope stability by melting snow or ice at higher rates at higher elevations (extreme temperatures), as well as enhancing thawing (mean temperature) (Huggel et al., 2012). In the long-term, increases in mean temperatures produce changes in the internal cohesion of steep slopes, allowing water percolation that enhances slope weakening and potential failures (Huggel et al., 2012). Heavy precipitation impacts, on the other hand, usually occur on short scales of time (less than a year) (Huggel et al., 2012). Additionally, (Tichavský et al., 2019) found that more significant sums of annual precipitation are related to landslides, which are enhanced by previous dry spells that can alter soil properties.

Landslide processes are particularly dangerous in areas close to human settlements: lakes that can overflow due to avalanches, as well as unstable valleys that, given the soil matrix and water content, can mobilize and produce mudflows (Carey et al., 2011; Haeberli et al., 2013). Therefore, the risk assessment of glacierized environments also needs to address the conditions downstream where landslides can trigger a chain of effects that increase the risk associated.

A study of rock glaciers in the Argentinean Patagonia (Drewes et al., 2018) found that there will be a dramatic shift upward of the 0 °C isotherm, changing on average 600 meters by 2070 following an RCP8.5 path. This is in the same order of changes in the permanently frozen ground line estimated (Saito et al., 2016) from the Last Glacial Maximum (21ka) to present.

In the Chilean Patagonia, studies are mainly carried out by the Sernageomin; some examples are (Sernageomin, 2003, 2008, 2011, 2018c, 2018b, 2018a). For example (Sernageomin, 2018a) estimated that in the Palena province alone, a small area compared to the Chilean Patagonia, there have been at least 2533 mass wasting events between 1965 and 2018, from which 713 correspond to landslides. However, there is a lack of understanding of the triggers and mechanisms that control such events and further studies need to be undertaken in order to understand the evolution of these events, linkages to climate change or anthropogenic changes, and to understand where they potentially can affect nearby villages, directly destroying houses and taking human lives (Gariano and Guzzetti, 2016) or indirectly affecting the connectivity of remote areas (Winter et al., 2016).



In this study, we want to explore the mechanisms that enable a landslide of $7 \times 10^6$ m$^3$ to evolve to the catastrophic
mudflow that hit Villa Santa Lucía in the Chilean Patagonia, resulting in 22 fatalities. The avalanche, which may
have been triggered by hydrometeorological conditions and destabilization of the wall around the receding Yelcho
glacier, led to the generation of a hyper-concentrated flow at the head of the Burritos River that traveled around ten
kilometers and affected 50% of the urban area of Villa Santa Lucia on December 16, 2017. The first hypothesis
indicated that the event was possible because of the presence of a glacier lake; however, there was no indication of
such features in the area. Therefore, this study seeks to understand the preconditions in the valley right below the
avalanche zone that contributed to this hyper-concentrated flow event and what conditions enabled this event
without the presence of a glacier lake, which will have a two-fold application. First, it will allow us to understand
the mechanisms of the chain of events leading to the 2017 mudflow in Villa Santa Lucia, and second, and probably
most important, to update the criteria for mapping risks associated with mudflows in the Chilean Patagonia.

## 2 Study Area

### 2.1 Location

Villa Santa Lucia is located in the valley of the Frio River, 75 kilometers south from Chaitén (closer town), along
the Carretera Austral, in the Los Lagos region, Chile. The avalanche event started at the head of the Burritos river
basin (43.413°S, 72.367°W) that runs to the west of Villa Santa Lucía (Figure 1). It begins at the eastern slope of the
western side of the regional Andes.

This area was under tectonic modeling associated with the Liquiñe-Ofqui Fault System (LOFS), forming valleys
with a NS-trend, such as the Frio River Valley and Yelcho Lake, which have been eroded by glacial processes in the
Pleistocene and, subsequently, filled by volcanic, alluvial and river processes (Sernageomin, 2018c). The climate of
the area presents intense thermal variations, high summer temperatures, and freezing temperatures in the winter
period; the rainfall reaches 3,000 mm annually, decreasing to the east (CECS, 2017).


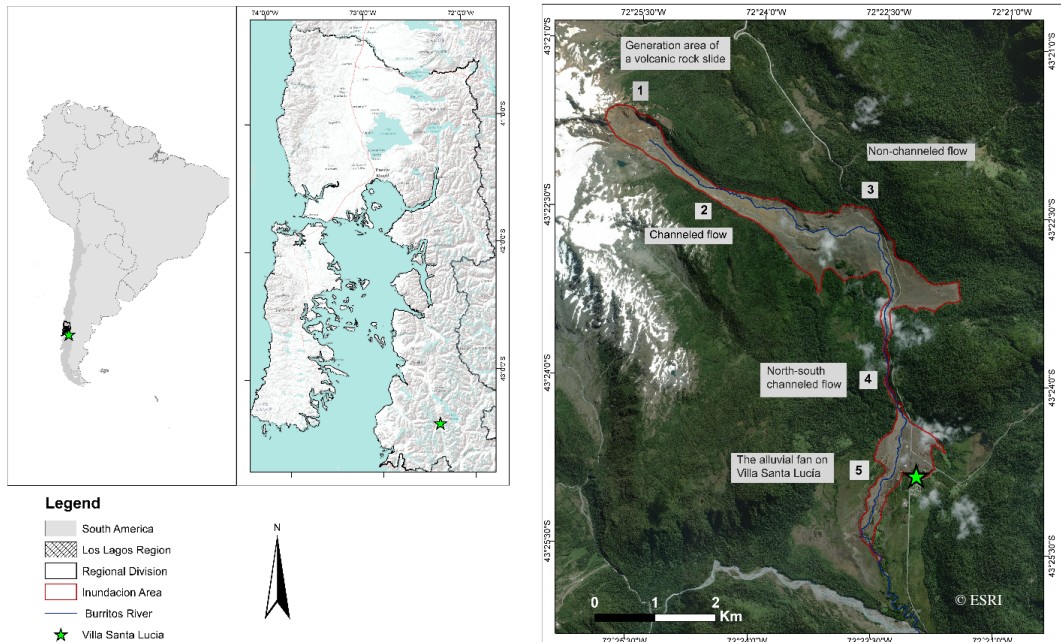

Figure 1:    Study area and extension of the inundation (South America and Los Lagos Region layer from https://tapiquen-sig.jimdo.com, Burritos River blue line layer from http://datos.cedeus.cl/, background © ESRI).

## 2.2  Geological setting

The study area consists mainly of 9 geological units (Figure 2). The dominant geological unit corresponds to an intrusive rock (Cretaceous and Miocene age) composed of tonalites, granodiorites, granites, diorites, among others. Moreover, the basement material is an old metamorphic rock. This unit is composed of micaceous shales, amphibolites, mafic, and ultramafic rocks. Volcanic and volcanoclastic rocks represent, in part, the NO-SE volcanic

cord, called the Cordon Yelcho Volcanic Complex (Sernageomin, 1995). While sedimentary rocks, mainly sandstone, shales, conglomerates, are presented as intercalations. Recent sedimentary deposits are mainly associated with rivers, alluvial, colluvial, morenic, and glaciolacustrine deposits (Aguilera et al., 2014) (Figure 2).

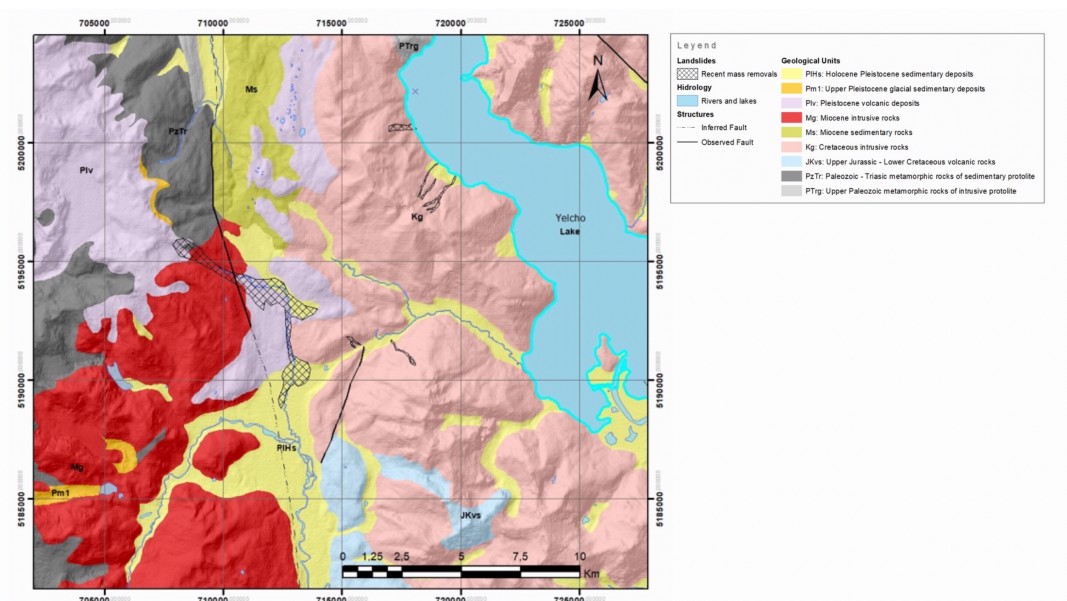

Figure 2:   Geology of the study area (Modified from Sernageomin (1995))

## 2.3  Event Background

The event in Villa Santa Lucía was triggered during a rainfall front passed over the area, in a year where the entire Los Lagos Region had had a significant surplus of rainfall, thus constituting a condition of soil saturation favorable to the occurrence of geohazards (Nguyen et al., 2018).

According to the information provided by the "Dirección General de Aguas" (DGA), at the time of the event, the total annual rainfall was 3,650 mm, and in the 30 hours prior to the event the rainfall reached 124.8 mm, with a maximum intensity of 10.6 mm/hr at 16:00 hours on December 15, 2017. This hydrometeorological event exceeds 99% of the historical precipitation events in the area (CECS, 2017). The precipitation events in Villa Santa Lucía occurred after two weeks with maximum daily temperatures that exceeded 22°C in at least nine days between

December 1st and 15th, with day 2nd and 5th of December being the hottest days with high temperatures that exceeded 27°C. During December, before the event, the air temperature reached an average of 14.9°C (CECs, 2017). The Center of Scientific Studies (CECs) determined through the data of a radiosonde located in Puerto Montt-El Tepual (-41.439, -73.094) that the isothermal level on December 15 was at 2,771 meters above sea level (m a.s.l.) for the Villa Santa Lucía coordinates. This implies that in the study area, there were only liquid precipitations in the days

before the event, even in the glaciers of the Yelcho range, considering that the maximum heights of the peaks do not exceed 1,800 m a.s.l. approximately.

On a report prepared after a volcanic event in the Chaitén Volcano in 2008 to evaluate the possible relocation of the town of Chaitén (Sernageomin, 2008) is concluded that "The surroundings of the Villa Santa Lucia presents a high and moderate danger of being affected by mass wasting processes, highlighting possible debris flows at the bottom

of the valleys and active channels. Areas that are not classified in high or moderate danger have low or no danger,

which should be assessed at greater detail. High flood hazard would affect only the flood plain adjacent to Villa Santa Lucia, although some degree of danger in the town itself must be evaluated on a more detailed scale." This shows that the area of Villa Santa Lucía and its surroundings were and are prone to mass wasting phenomena. Although the town itself was not in danger, it was mentioned that more detailed studies were needed in the area.

Sernageomin (2018b) compile a series of reports; several studies are mentioned that help to understand the origin and explain the event as a whole. On December 16, 2017, a 7 million cubic meter volcanic rock slide detached (Figure 3, point 1) falling on the Yelcho glacier. The glacier tongue sits on an intrusive formation that has a drop of about 80 meters. The glacier has been steadily retreating during the last decade. Sernageomin (2018c) indicated that there might have been a small lake and water available under the glacier at the terminal, but there is no conclusive

information. We checked satellite images before the events, and there is no indication of a lake above the intrusive unit. Two million cubic meters of the material stayed above the intrusive, and five million cubic meters of material continue downstream, sliding on the intrusive unit at an angle of more than 70 degrees. At this point the material encounter the Burrito River that has high slopes, so the detritus flow continued along the river at high velocity (Figure 1 and 3 point 2) scouring the walls of the river and adding a significant amount of sediment to the

avalanche. The sediments are mainly associated with glaciolacustrine deposits (easy to mobilize) and ancient alluvials present in the valley and in the river walls.

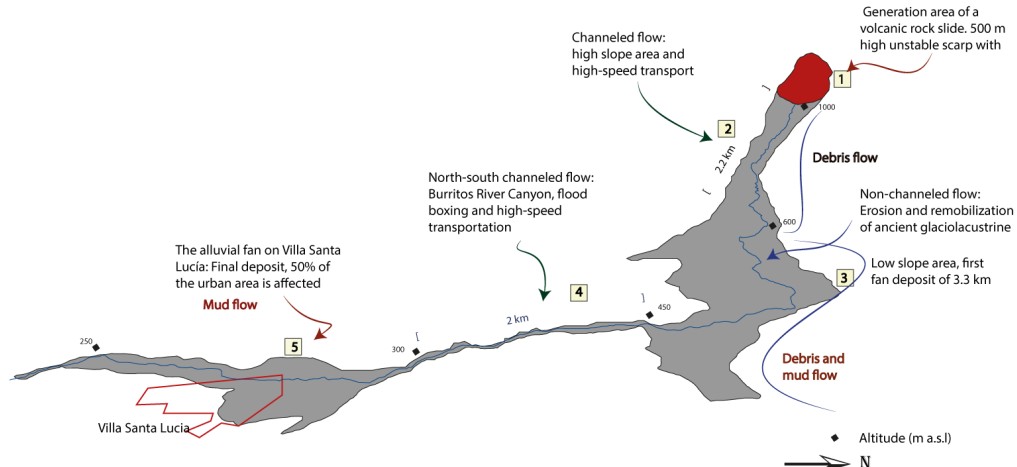

Figure 3: Flow trajectory scheme and details of the five key sections identified by Sernageomin and our fieldwork.

Additionally, the soil was almost saturated, which added water to the mix, transforming the avalanche flow into a

mudflow. Moreover, dense forest was present, which added a significant amount of biomass to the mix. Then the mudflow reaches an area with low slopes at a distance of 8.6 km from its origin through the Burritos River to the east. In its trajectory, the mudflow crossed Route 7 (Carretera Austral) in a stretch estimated of 2 km and filled an old wetland. In that sector, the flow was channeled in a canyon oriented north-south toward Villa Santa Lucia in a section of 1.5 km (Figure 1 and 3, point 4). Once the flood reached Villa Santa Lucia, it slowed down and deposited

the sediment in a radial fan of 600 to 1,000 m with a height of 1 to 5 m (Figure 1 and 3, point 5).



Through field observations, we identified five key sections that describe what happened in Villa Santa Lucía (Figure 1 and 3).

1.   Generation area: It has a range of peaks between 1,000 to 1,400 m a.s.l., with the main escarpment of 900 m
long and 520 m wide. The north wall has maximum slopes between 77 to 81° (Figure 3 and 4)

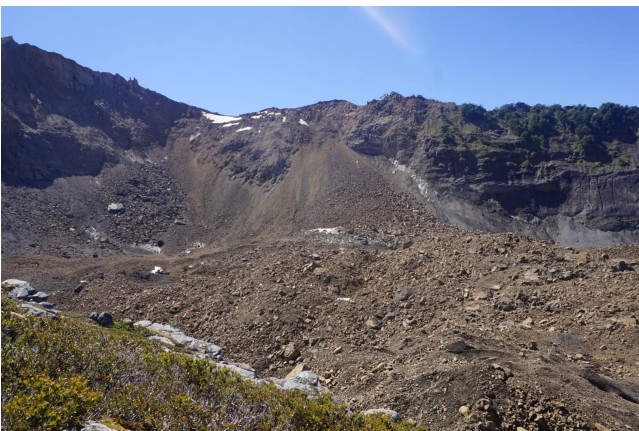

Figure 4:   Area of slope failure slid and deposit.

As a result of the avalanche, the material deposited acts as a dam creating two small lakes. The largest lake has a length of 180 m and an average width of 50 m (Sernageomin, 2018c).


2.   Channeled flow: Corresponds to the upper segment at the foothills, the flow is channeled and traveled 2 km. The flow width was between 200 to 400 m wide. Using photo-interpretation of the digital elevation models (SAF, 2017) and aerial photos, Sernageomin personnel (Sernageomin, 2018c) estimated an approximate wave of 20 m height (Figure 3 and 5).

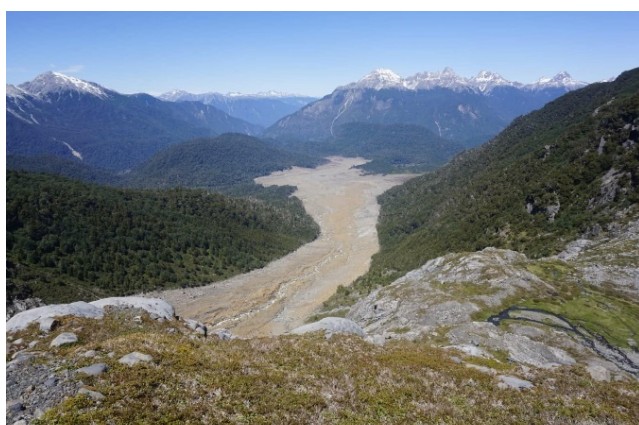


Figure 5:   Channeled flow at the foothills.



3. Non-channeled flow: This section has a length of 2.3 km in a west to east direction until the Burritos river changes its orientation to a north-south direction. The elevation goes from 600 to 380 m a.s.l. The slopes decrease slowing down the flow, which deposited sediments. The non-channeled flow width reaches 1.4 km. In this sector, the flow went over the road (Carretera Austral) to the east in 1.3 km affecting an old wetland (Figure 3 and 6).

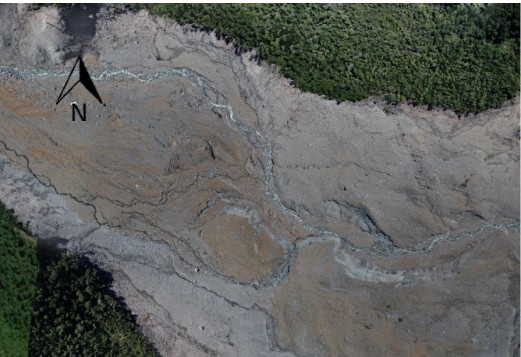 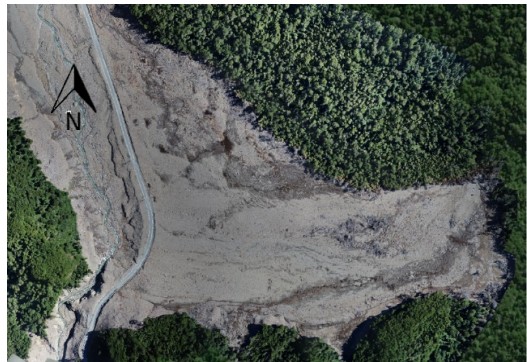

Figure 6: Aerial photo of non-channeled flood deposited in an old wetland captured with an InspireII UAV.

4. North-south channeled flow: The flow descends through the enclosed channel at the Burritos river canyon from 380 to 250 a.m.s.l. on a 2 km long section (Figure 3 and 7).

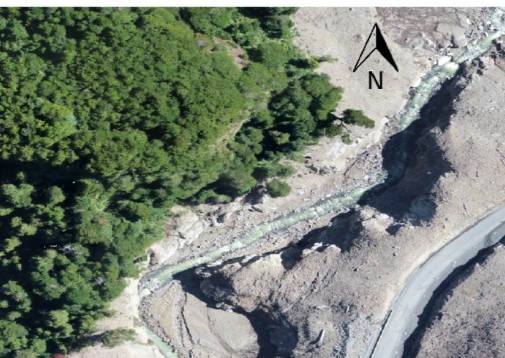 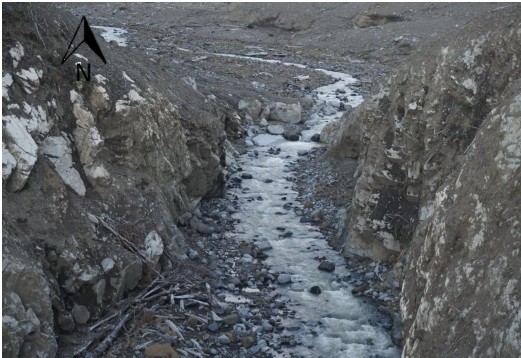

Figure 7: Aerial photo of the channel in the last section before entering Villa Santa Lucia (left) captured with an InspireII UAV. Picture of the channel facing downstream (right).

5. The alluvial fan on Villa Santa Lucía: due to the lost in confinement the flow expands with a radial distribution in a southeast-southwest direction of 600 to 1,000 m (Figure 3 and 8)





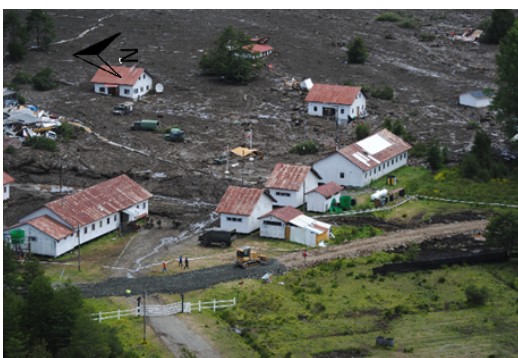
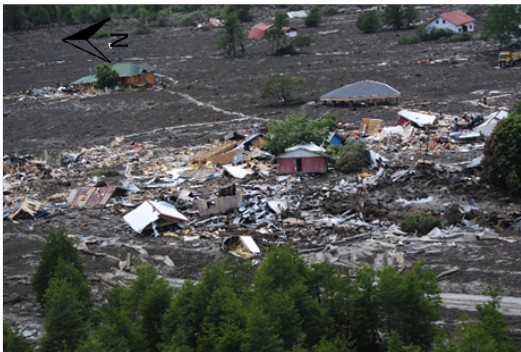

Figure 8: Villa Santa Lucia after the mudflow (Used with permission from Sernageomin (2018c)).

Also, Sernageomin (2018c) estimated the volume of material mobilized during the event using digital elevation

models (DEM) created from photointerpretation of the orthomosaic provided by SAF (2017). Sernageomin (2018c) identifies the release and the deposition area at the intrusive formation before the mudflow started. They compared digital elevation models before the event, Intermap, and SRTM 30, with the digital elevation model created by SAF (2017) The avalanche volume was on the order of 7,200,000 m³. Likewise, they estimated that approximately 2,200,000 m³ of sediments were deposited in the upper part of the basin and that approximately 5,000,000 m³ were

the contribution to the flow.

The speed of the flow at the Burritos river canyon (Figure 6) and at the beginning of Villa Santa Lucía when the flow opens (Figure 7) is approximately 20 m s⁻¹ (Sernageomin, 2018c)

## 3 Methodology

### 3.1 Fieldwork and Geotechnical sampling

A fieldwork campaign was carried out in January 2019 to map in high resolution the area using an unmanned aerial vehicle (UAV). We used aerial photogrammetry to produce a high-resolution DEM for some parts of the study area. Aerial photogrammetry allows creating 3D models from 2D images, obtaining geometric characteristics of the objects they represent. We used a UAV Inspire II with a Zenmuse X4 camera to capture aerial images (Figure 9). We did not have differential GPS at the moment of this survey to take control points to correct the DEM generated.

Therefore, we just used this DEM to corroborate that the DEMs with lower resolution and freely available were able to capture specific features such as canyons or small changes in slopes that could affect the hydrodynamic simulation and the path of the flood. We also generated a high resolution mosaic to observe details of the flood. We used the software Agisoft PhotoScan Professional 1.4 desktop to process the images. The software performs the restitution of images by spatial coincidence between the elements represented in each image.




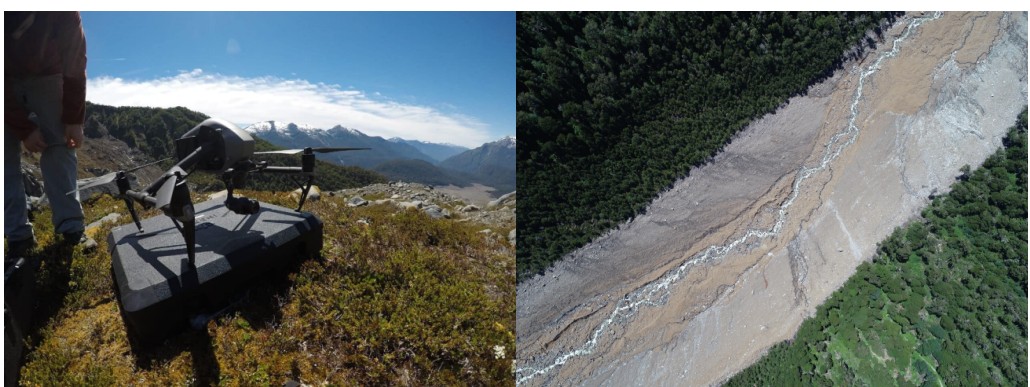

Figure 9:   UAV mapping survey

Additionally, we extracted an intact soil sample (Figure 10) to determine the physical and mechanical parameters
extracted from an undisturbed sample following the American Society of Testing Materials ASTM D3080 (ASTM,
2017a).

The tests performed to determine the geotechnical properties of the primary materials were: direct shear test,
unconfined compression, density, and soil classification. The parameters obtained in each laboratory test will serve
as constrains for the numerical modeling of the hyper-concentrated flow of Villa Santa Lucía.


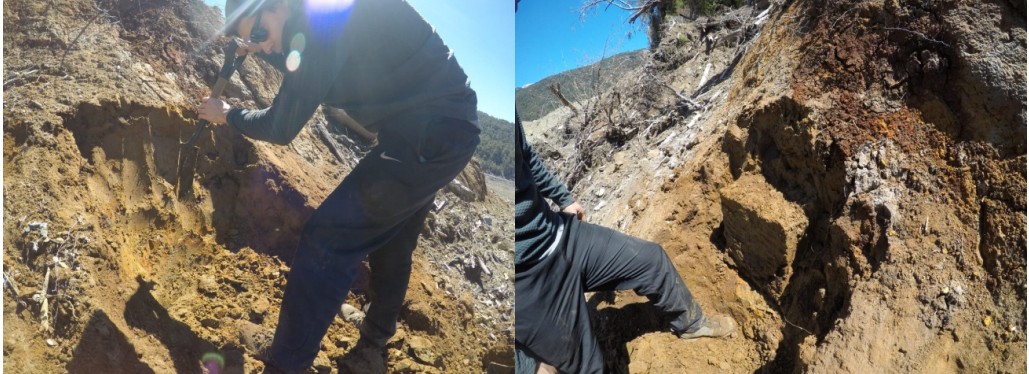

Figure 10: Extraction of an undisturbed soil sample

## 3.2  Numerical Modeling

The modeling of the hyper-concentrated flow is carried out using two software: r-avaflow and Flo-2D, which we
describe below:

### 3.2.1   R-Avaflow



R.avaflow is a free computer software that offers a practical, innovative, and unified solution to simulate granular and debris flows produced in high mountains around the world. The model can handle rapid mass flows, including avalanches and two-phase flows (Mergili, M., Pudasaini, 2019). R.avaflow calculates the propagation of mass flows

from one or more given release areas on a defined baseline topography until (I) all material has been stopped and deposited; (II) all the material has left the area of interest; or (III) the maximum simulation time has been reached. R.avaflow is developed in two formats for its environment and operation, r.avaflow expert and r.avaflow professional. The latter represents an autonomous version with reduced functionalities. It operates through a graphical user interface (GUI). In this study, we used the professional version (Mergili et al., 2017). This model

represents a complete open-source computational framework based on a geographic information system (GIS) that offers a two-phase flow model. Also, consider the drag of material along the flow path. These characteristics facilitate the simulation of complex mass flows, as well as chained processes and interactions.

For the propagation of the flow, we used the Pudasaini model (Pudasaini, 2012), which is a two-phase mass flow model. Solids and fluids materials can be dragged from the bottom and incorporated into the flow. The r.avaflow

output consists mostly of raster maps of solid and fluid flow heights, velocities, pressures, kinetic energies, and dragged heights.

### 3.2.2   Flo2D

Flo-2D is a two-dimensional finite differences model (FLO-2D, 2018; O'Brien and Zhao, 2004) that can simulate non-Newtonian flow. The model allows simulations in complex topographies, urbanized areas, and floodplains, allowing fluid exchange between the channels and the floodplain. It can model water flows, hyper-concentrated

sediment flows, and mudflows. The input data required are a digital topography of the land, the geometry of the channel, estimated values of the channel roughness and the flood plain, liquids and solids inputs, precipitation, and rheological properties of the water-sediment mixture. The software calculates the surface flow in eight directions, considering the conservation of the mass, it uses a variable time step by increasing and decreasing the scheme that incorporates efficient numerical stability criteria (FLO-2D, 2018). The rheological model used in FLO-2D is based

on the work of (Julien and Lan, 2007; O'Brien et al., 1993; O'Brien and Julien, 1988) which describes the dynamic viscosity and the shear stress os the mixture as an exponential function of the sediment content (FLO-2D, 2018).

### 3.2.3   Tography

In the study area, we have two freely available DEMs, the Shuttle Radar Topography Mission DEM (SRTM) of 30

meters resolution and the ALOS-PALSAR DEM of 12.5 meters resolution per pixel. Alganci et al. (2018) and Caglar et al., (2018) compare these two DEM and others global procucts concluding that ALOS-PALSAR has lower errors. Therefore, we used it for numerical simulation. We resampled the resolution of ALOS PALSAR to 30 meters to speed up the simulations.

### 3.2.4   Models calibration

For the calibration of the models, we used three criteria (1) Flood area, which was mapped after the event using aerial imagery (SAF, 2017); (2) Flow heights estimated by Sernageomin in Villa Santa Lucía and at the beginning of





the canyon; and (3) Flow velocities in the canyon curve and at the beginning of Villa Santa Lucia (Figure 11). See Table 1 for the values.

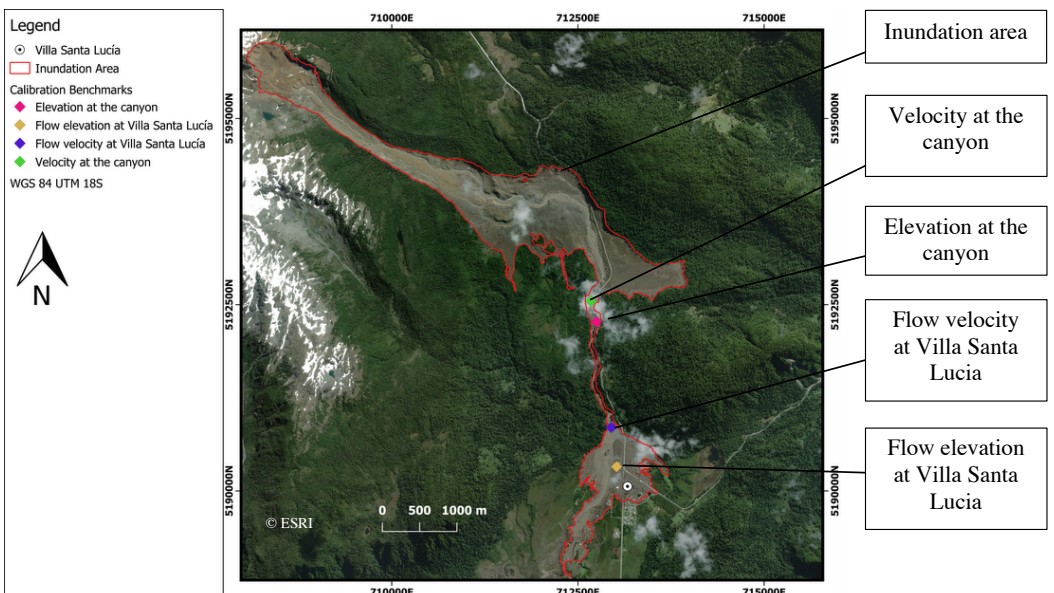

Figure 11:  Location of the parameter estimation for the calibration of the models (Background © ESRI)

Table 1: Parameter for the calibration

| Parameters | Magnitude | Units |
|---|---|---|
| Inundation area | 4,926,533 | $m^2$ |
| Elevation at the canyon | 20 | $m$ |
| Flow elevation at Villa Santa Lucia | 2 | $m$ |
| Velocity at the canyon | 20 | $m\ s^{-1}$ |
| Flow velocity at Villa Santa Lucia | 21.42 | $m\ s^{-1}$ |

To define the model that best adjusts to the Sernageomin's estimations,  we calculated the standard deviation from the modeling results and the parameters from Table 1.  The parameterization of the model with less standard deviation is considered the best parameterization for the particular software used.


## 4  Results

### 4.1  Geotechnical results

It was not possible to carry out the extraction of unaltered geotechnical samples from the upper part, mainly due to the 5 kilometers of steep terrain, including a section that needed to be climbed to transport the material by foot.

However, it was possible to collect soil and rocks, which, although not in an unaltered condition, help to characterize the soil that originated the hyper-concentrated flow. We extracted soil samples in the middle-low part of




the first section of the flow. This soil sample represents the material added to the flow; thus, its properties become very important when it comes to reproducing the hyper-concentrated flow of Villa Santa Lucía.

For the soil moisture, a wet portion of the soil was extracted and allowed to dry in an oven at 60°C for two days.

With this, it was possible to calculate mass soil water content (by weight) in 1.96 $g_{water}$/$g_{solid}$

Then, a fraction of the soil sample was covered with paraffin to seal all pores and thus determine its density through the submerged weight difference. The wet soil density is 1.24 grams per $cm^{3,}$ and the dry soil's apparent density is 0.419 grams per $cm^3$ — the specific weight Gs=2.65 gr. Therefore, the void ratio "e" is 5.324. Using these values, it turned out that the soil porosity is 84.18%, and the water saturation of the soil sample 97.4%; therefore, considering

that the total porosity is 84.18%, 81.95% of the total unaltered soil volume corresponded to water.

Then a direct shear test was performed in three probes. Each probe was loaded with 2, 4, and 8 kg, respectively, allowing them to consolidate for 24 hours. We recorded the deformations versus time in the first sample to determine the speed for the direct shear test using the square root scale method (Table 2).

Table 2: Direct shear test results

| Square root scale method | |
| --- | --- |
| $t_{90}$ [min] | 1.44 |
| Conversion factor from $t_{90}$ $to$ $t_{50}$ [min] | 0.34 |
| Total time estimated [mm] | 16.70 |
| Displacement estimated [mm] | 10.00 |
| Displacemente rate obteained [mm/min] | 0.599 |
| Displacement rate to use  [mm/min] | 0.5 |

With these results, we estimated a cohesion value of 22.5 kPa and an internal friction angle of 23.8°.

**4.2  Soil Classification**

In order to carry out the classification of the soil, we used two tests. First, we determined soil´s granulometry using the American Society of Testing Materials ASTM D2487 (ASTM, 2017a), and second, we determined soil consistency limits (liquid and plastic) using the American Society of Testing Materials ASTM D4318-17e1 (ASTM,

2017b). For the granulometry, Table 3 shows what percentages pass through the different sieves' sizes. We omitted the larger sieves since 100% of the material passed throughout them.

Table 3: Percentage that passes through sieves

| Sieve size | Percentage that passes |
| --- | --- |
| Sieves 4.75 mm | 100 |
| Sieves  2.0 mm | 99 |
| Sieves 0.425 mm | 96 |
| Sieves 0.075 mm | 73 |





Finally, the liquid and plastic limits of the soil are 50% and 27%, respectively. According to the USCS

classification, the soil corresponds to a CH, an inorganic clay of high plasticity, while for the classification of the

AASHTO, it is considered to be clay.

### 4.3  Numerical modeling

According to the antecedents of the event, the avalanche fell on the glacier terminus. A fraction of the material

stayed deposit between the glacier and the intrusive formation. The difference (about five million cubic meters of

sediment) continued downstream, initiating the scouring process and mudflow that lead to the Santa Lucia event.

Therefore, the models' domain starts at the base of the intrusive formation with an initial volume of 5.000.000 cubic

meters. For the calibration, we matched the observations and the empirical calculations by Sernagomin of the

velocities and depositions.

### 4.4  R-Avaflow

First of all, in order to simplify the calibration, we divided the process into two. First, we set the sediment

concentration by volume of the flood in 50% and change the drag coefficient, basal friction angle, environmental

resistant coefficient, and fluid friction coefficient. Table 4 shows the first set of parameters used.

Table 4: Initial parameters for the r-avaflow simulations

| Solid density [$g/cm^3$] | 2.400 | Terminal Velocity | 1 |
|---|---|---|---|
| Liquid density [$g/cm^3$] | 1.000 | Contribution parameter S-L drag resistance | 0.500 |
| Virtual mass | 0.500 | Fluid friction coefficient | 0.002 |
| Hydrograph | No | Output writing time (s) | 10 |
| Diffusion control | Si | Internal friction angle | 24 |
| Conservation of volume | Si | Particle Reynolds number | 1 |
| Surface control | Si | Exponent for drag | 1 |
| Viscous shear coefficient of the fluid | 0 | Quasi Reynolds Number | 4.500 |
| Solids concentration distribution with depth | 0 | Mobility Number | 3 |

The best set of parameters is 5.75, 2, 0.022 and. 0.0005 for the drag coefficient, base friction angle, environmental

resistant coefficient, and fluid friction coefficient, respectively, and the corresponding map result is in Figure 12.

high

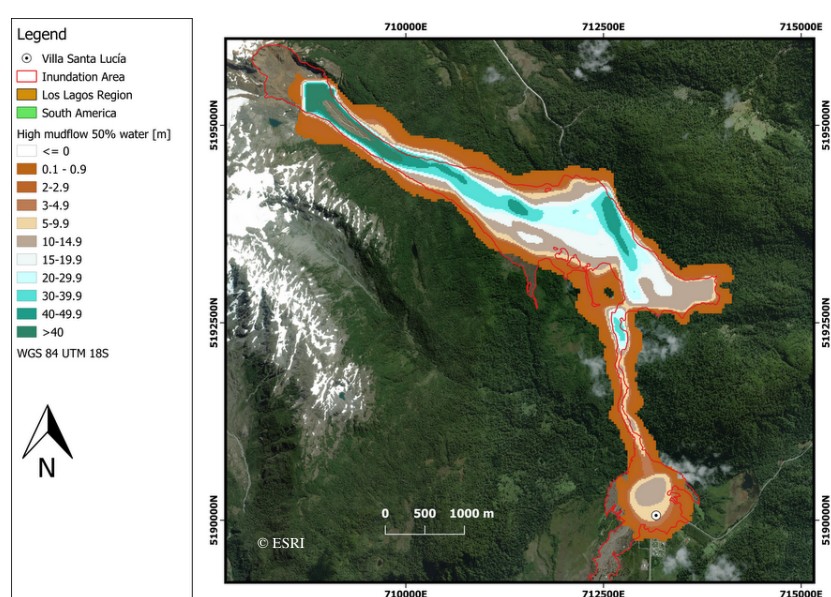

Figure 12:        R-avaflow modeling results for a concentration by volume of 50%.

Figure 12 shows good agreement with what was reported by (Sernageomin, 2018c). However, the mixture is still

largely fluidized, and it does not follow the edges of the inundation. For this reason, we performed simulations with

the same input parameters but a varying percentage of initial water. We changed the percentage of water from 20%

to 70%. The error for the heights, speeds, and pressures calculated in each model are in Figure 13.

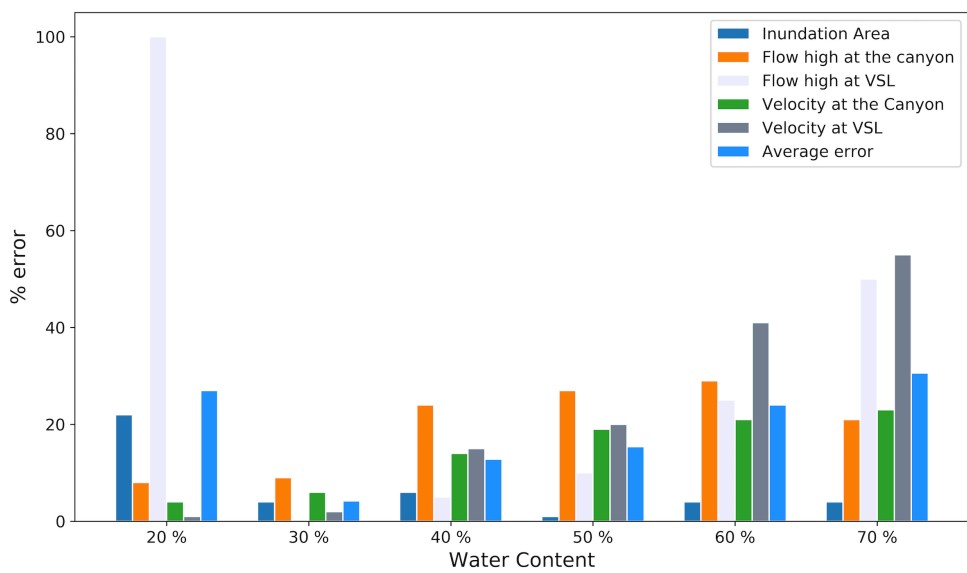

Figure 13: Error obtained with different water content and the best set of parameters obtained in the first calibration.





From Figure 13, we concluded that a mudflow with a volume of water of 30% could reproduce best the VSL event (Figure 14).

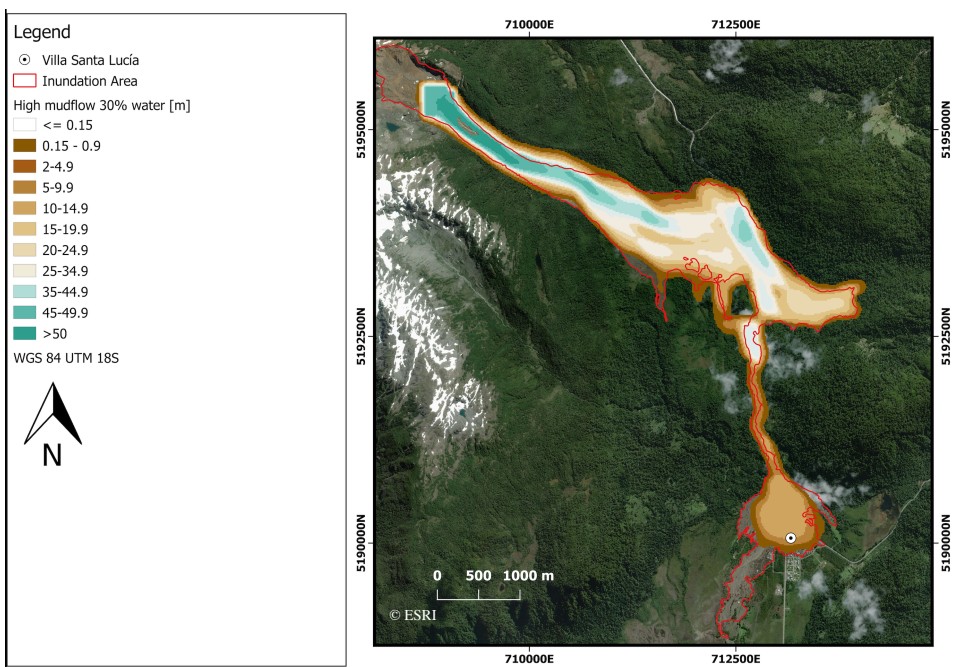

Figure 14: Best simulation results using the calibrated parameters for r-avaflow and water content of 30%
(Background © ESRI)

### 4.5  Flo2d

The main results from Flo-2D for the hyperconcentrated flow in Villa Santa Lucía are presented below. The table below shows the parameters used for modeling.

For the laminar resistance k, we used the default number 3000. The influence of the value of $K$ does not affect
simulations significantly compared to other parameters related to flow resistance (Hsu et al., 2010). The specific gravity of sediments Gs is equal to 2.65 g cm⁻³. For the sediment concentration by volume, we used values between 40% to 50%, which correspond to hyper-concentrated flows. The error for the heights, speeds, and pressures calculated in each model are in Figure 15. Moreover, Figure 16 shows the extension of the best model using Flo2D.

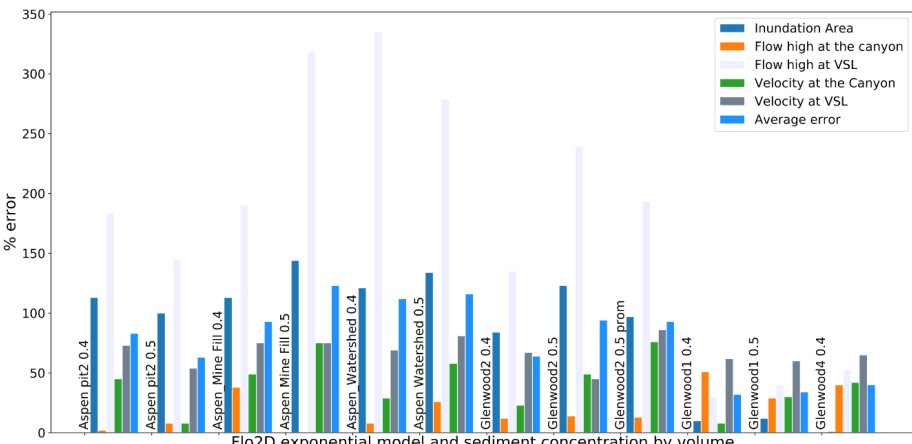

Figure 15: Sensitivity analysis for different dynamic viscosity and shear stress models (FLO-2D, 2018)

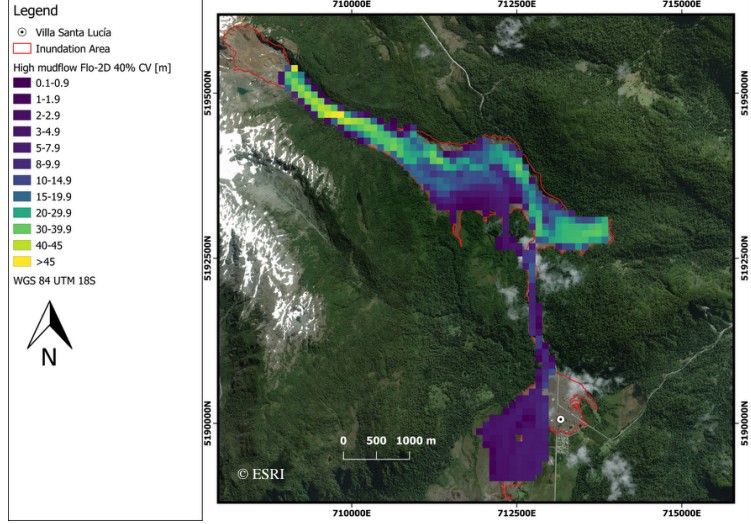

Figure 16: Result for Glenwood 1 using a 40% concentration by volume (Background © ESRI)

## 5 Discussion

### 5.1 Geotechnical sample

The soil was classified as clay with high plasticity (CH), with a friction angle of 23.8 ° and a cohesion value of 22.5 kPa, which are characteristic of a consolidated clay. However, the soil extracted had various types of soils and minerals in its composition, which made this a rare soil that is not described or classified by USCS.

For the granulometry, 72% of the soil passed through sieve No. 200. Mainly sand and volcanic soils did not pass sieve No 200. Similar results were founded by Gonzalez-Pulgar (2012) in volcanic soils that have an internal soil

friction angle of 25° and a cohesion value of 2.9 kPa. Moreover, our results were consistent with their high water


content by soil, greater than 150%, and a very low dry density of solids with values between 0.4 and 0.7 kg cm$^{-3}$. The values obtained by (Gonzalez-Pulgar, 2012) are comparable to the results of this study.

The results of the dry density and natural moisture determination tests help to know the natural state of the unaltered soil sample obtained in the flood zone of the hyper-concentrated flow of Villa Santa Lucía. Besides, these tests help

to calculate intrinsic soil parameters, such as the relative density of solid particles (2.65 g cm$^{-3}$), the void ratio, the degree of saturation, and porosity. The humidity estimated under ASTM D2216-19 (ASTM, 2019) showed a value of 195.85% and a dry soil density of 48.1 kPa. The soil saturation was 97.4%. Therefore, 81.58% of the soil volume was water.

The previous result is relevant because the liquid and plastic limits are 50 and 27%, respectively. These parameters

indicate if the soil has a more liquid or plastic behavior, given the moisture content. Also, incorporating the moisture of the soil sample of 195.85%, it is possible to determine the liquidity index, resulting in a value of 7. When the liquid index is higher than one, and the soil is subjected to cutting, it has a viscous liquid behavior. Such soils can be susceptible to the collapse of the soil structure. As long as they are not altered, they can be relatively stable, but if these soils are subjected to shearing (molding), and the soil structure collapses, then they can easily flow like a

viscous liquid (Holtz and Kovacs, 1981). These results show that the area downstream of the collapsed wall was prone to produce a mudflow due to the high water content and the perturbation produced by the avalanche mobilize the soil bed producing the hyper-concentrated flow that affected Villa Santa Lucía.

### 5.2 Numerical Modeling

The modeling in the r.avaflow software successfully reproduced the hyper-concentrated flow of Villa Santa Lucía.

The model was able to replicate the extension of the floodplain, its critical heights, and speeds at the points reported by (Sernageomin, 2018c). We run three hundred simulations in order to calibrate the model. From these simulations, it was possible to determine that the general drag coefficient is the most critical of all since even a change of a decimal in the coefficient changed the viscosity of the flow, reducing or exaggerating the flood runout. The sensitivity analysis shows that r.avaflow is also sensitive to changes in the basal friction angle. These strongly

condition the rheology of the flow, determining the height, and velocities.

The results obtained in the modeling in the FLO-2D software were diverse. We could not achieve the same level of r-avaflow's results. The inundation area was the most sensitive result in the concentration of sediments by volume and the rheology model (Figure 15). The best combination of parameters was the Glenwood 1 rheology model and a 40 % concentration sediment, which has an error of 32%.

From the numerical modeling, we concluded that we could reproduce the mudflow satisfactorily using r-avaflow with water content in the mixture ranging from 30 to 40 %. (Figure 15)

### 5.3 Source of water

From the observation and information provided by Sernageomin, our soil samples analyses, and numerical simulation, we found that the primary source of water was in the soil. With r.avaflow, we estimated that mudflow

scoured 3,402,100 m$^3$ from the Rio Burritos, which were added to the original 5,000,000 m$^3$ from the avalanche. The soil was saturated at the moment of the avalanche due to the intense precipitations and soil properties. Considering that the soil porosity is 84.18% and assuming a saturation of 97.4% the same as what we found during our


fieldwork, there were 2,789,500 m$^3$ of water within the soil scoured by the event. Additionally, extra water was added, not quantified in this study, from the ice pieces detached from the glacier and carried by the avalanche first and mudflow later.

Our results highlight the importance of considering the potential chain of events in the risk analysis and hazard mapping. In practice to foresee a mudflow hazard, we seek areas where evident water sources are available such as glacial lakes. However, in this work, we showed that it is possible to have the catastrophic event that hit Villa Santa Lucia just with the water within the soil in the valleys downstream of the avalanche. Additionally, the soil has a particular volcanic origin, and it is highly fluidizable under low lateral effort. This result is relevant for the Patagonia region because, in order to update the hazard maps, we need to consider not only the primary events but also the potential reactions from the downstream elements of the chain of events, which as in this case, may not be evident previous the event.

## 6 Conclusions

The combination of geotechnical tests and freely available computational software to model mudflow are useful tools to characterize and reproduce mass wasting events, such as the mudflow occurred on Decembre 2017 in Villa Santa Lucia in Chile. Our results present the possibility of open-source software implementation to represent mudflow events in the Patagonian Andes with a good nor better performance. These types of studies will allow integrating better methodologies to enhance risk scenarios related to mudflow events in active subduction zones like the Andes.

What actually triggered the rockslide event is not clear and not part of this research. We focused in what conditions in the valley enabled that the avalanche had evolved into mudflow and traveled all the way to Villa Santa Lucia. We have no information to guess how much water was available at the glaciar terminal so we wanted to understand if this event was possible without a lake or large water reservoir at the glacier. Our results show that in fact the water available around the Burritos River was sufficient to transform the detritus flow into a mudflow. Our numerical models and geotechnical work show that the mudflow event was possible by pre-existing water in the saturated soil. Geotechnical evidence demonstrates a high plasticity soil (CH) with a low internal friction angle associated with volcanic soils of Holocene age. Hence, weak soils prone to mass wasting events due to their geomechanical properties. Given the complexity and the potential increase in the future of extreme events occurrence that can trigger mass wasting, we suggest that hazard studies should consider the structural conditions that are present in the area of influence of the mudflows. Soil characteristics ough to be included because they may play a crucial factor amplifying the impacts of local events trigger by hydroclimatic events, which is the case of Villa Santa Lucia, where the water necessary to fluidize the mudflow mixture was in the soil in a volatile system that was easy to mobilize.



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
