# Peer review of "The mudflow disaster at Villa Santa Lucía in Chilean Patagonia: understandings and insights derived from numerical simulation and post-event field surveys"

_Natural Hazards and Earth System Sciences, 2019_

## Referee Comment (RC1) · Martin Mergili (Referee) · 28 Mar 2020

Review to the paper

**Hidden Hazards: the conditions that potentially enabled the mudflow disaster at Villa Santa Lucía in Chilean Patagonia**

Submitted to NHESS by *Marcelo A. Somos-Valenzuela* et al.

Reviewer: *Martin Mergili*

The authors describe a cascading landslide event in the Andes of southern Chile. Besides an analysis of the general situation and of geotechnical samples, a particular focus is put on the numerical simulation of the mud flow involved in the process chain: thereby, the authors use and compare the software tools FLO-2D and r.avaflow to back-calculate the propagation of this flow-like event. The work represents an important case study, that adds another piece to the hard-to-solve puzzle of understanding complex landslide events in changing mountain areas. As such, I would like to see this paper published. Before, I recommend some **major revisions** which mainly concern the consistency between title/abstract/introduction/conclusions and the main content, and the clarity of some explanations. Please find my general and specific comments below.

**General comments**

1. Even though the paper is mostly written in an understandable way, there are various language issues which require substantial polishing (if possible, by a native speaker).

2. Title: the title sounds nice, but does it really describe the main content of the paper? I have the feeling that the paper is not so much about the conditions that potentially enabled the mud flow disaster, but rather on the propagation of the mud flow.

3. Abstract: in my opinion, it could be condensed and a little bit more focused, but this is an issue of preference. The general statements at the beginning are maybe too long, given that the paper mainly describes a case study.

4. Introduction: the same as for the abstract. It starts with "Climate change", and the first three paragraphs deal with climate change-landslide relations. Even though this is certainly an important topic, it is not the subject of the paper when looking at the methods, results, discussion and conclusions. If the authors would like to stay with the focus on climate change, this aspect has to be more strongly included in the main content of the paper. Otherwise, the introduction should be restructured and reformulated, shortening the part on climate change and coming more quickly to the core topic of the paper.

**Specific comments**

L36: Please mention already here the region (southern Chile), many readers might not know the Yelcho mountain range.

L47 (and in general): r-avaflow -> r.avaflow

L49-50: You cannot determine the total water content from simulations and soil tests – you can just estimate it. Further, the precision given in the volume number (also in some other places) is too high, considering the uncertainties. In this case, 2.8 million m³ would be sufficient.

L113: "… alluvial and river processes …": aren't alluvial processes also river processes?

L124 NO -> NW

Figure 2: Nice figure, but there are some hispanicisms is the legend … "Leyend" -> "legend"; "Hidrology" -> "Hydrology"; further, in the map itself the lake polygon should be deselected before exporting the map.

L136: What would be the average annual rainfall in this area?

L156: A new section about the description of the event should start here.

L157: I would expect a little bit more information on the landslide-glacier interaction. Was there some glacier ice entrained, which was included in the flow downstream? This can be an important issue, even though it is not always straightforward to analyze its importance (see e.g.: *Mergili, M., Jaboyedoff, M., Pullarello, J., Pudasaini, S.P. (2020): Back-calculation of the 2017 Piz Cengalo-Bondo landslide cascade with r.avaflow. Natural Hazards and Earth System Sciences 20: 505-520. doi:10.5194/nhess-20-505-2020*)

L175: Maybe add some brief information (1 sentence) about damages and casualties, readers might be interested in that. As the term "disaster" is included in the title of the paper, there should be at least some information on the socio-economic component.

Figs. 4-8: They are very informative, but I recommend to put them together into one full-page figure with 7 panes or so.

L273: "Topography". Further I recommend to shift the section 3.2.3 farther up, as it rather concerns data acquisition. In the place where it is now, it disturbs the flow of reading from the models to the parameterization.

Table 1: you may round the inundation area and the flow velocity at Villa Santa Lucía – the numbers indicate a precision which is probably not justified by the data.

L289: How did you perform the calibration? Did you just use an iterative optimization procedure ("trial and error"), or did you use some automated, systematic procedure? Please explain! These things are explained a little bit in Section 4 (results), but they should already be explained in the methods section.

4.3.: The heading "Numerical modelling" is misleading as, in this section, just the modelling domain and the calibration procedure are briefly described. This is something I would rather expect in the methods section, as it is not a result. Further: did you also consider simulating the entire event (including the initial landslide?) This could be an interesting task for the future and, as such, could be mentioned in the discussion. There is now the multi-phase model of Pudasaini (*Pudasaini, S.P., Mergili, M. (2019): A Multi-Phase Mass Flow Model. JGR Earth Surface. doi: 10.1029/2019JF005204*), which could also serve for the simulation of the interaction between the landslide and the glacier.

Section 4.4/Table 4: Again, some Hispanicisms (Si->Yes). Further, there is no information about entrainment. Did you allow entrainment and, if yes, which value did you set for the entrainment coefficient? The "environmental resistance coefficient" is the "ambient drag coefficient", I think. Further, the Quasi Reynolds number and mobility number are 10^4.5 and 10^3. It is the logarithms which are given in the r.avaflow input.

Fig. 12 (and some other places): flow high -> flow height

L341/342: Better: "… We varied the percentage of water between 20% and 70% …" – the formulation as it is now is misleading.

Fig. 14, legend: revise the thresholds: e.g., to which class would a flow height of 49.95 m belong?

Section 5.2.: You should also briefly mention the limitations of your calibration due to issues of equifinality (e.g. *Beven, K. (1996). 12 Equifinality and uncertainty in geomorphological modelling. In The Scientific Nature of Geomorphology: Proceedings of the 27th Binghamton Symposium in Geomorphology, Held 27–29 September 1996 (Vol. 27). John Wiley & Sons.*), and the multi-dimensional parameter space (e.g. *Saltelli, A., & Annoni, P. (2010). How to avoid a perfunctory sensitivity analysis. Environmental Modelling & Software, 25(12), 1508-1517.*).

L400: The water content leading to the empirically most adequate results was approx. 30%. Is this fraction also plausible from a physical point of view, and from the observations? Please briefly elaborate on this aspect more explicitly (some indirect information is given in the paragraph below).

L404, 405: Oh, you computed entrainment! This is good, but it is mentioned here for the first time (unless I overlooked it). You should appropriately address this important aspect also in the methods and the results sections.

L409: Now, the entrainment of glacier ice comes into play! You have to introduce this aspect already in the event description (see comment above).

L420: FLO-2D is not a freely available software.

Conclusions, second paragraph: I would rather suggest to move this text to the discussion. The conclusions should rather focus on the key messages from your work (just extend what is written in the first paragraph).

This is all from my side. If the authors disagree with the one or the other comment, or would like to discuss issues, they should feel free to contact me at martin.mergili@univie.ac.at.

With best regards

Martin Mergili

---

## Referee Comment (RC2) · Haruyuki Hashimoto (Referee) · 26 Apr 2020

Catastrophic landslide and debris flow is an important topic in the fields of geology, geomorphology and civil engineering. This paper deals with an example of the catastrophic landslide and debris flow event in Chile. Therefore, the reviewer thinks the present paper valuable. The referee comments are as follows:

1. Pages 2 to 20: There are various technical terms expressing 'sediment-related disaster', such as rockslide, landslide, avalanche, hyper-concentrated flow, hyper-concentrated sediment flow, mudflow, debris flow, debris and mud flow, detritus flow, avalanche flow, mass flow, two-phase flow, non-Newtonian flow, and water-sediment

[Figure]

mixture. These similar technical terms make us confused. In order to avoid the confusion, the authors should unify these similar words and then describe the definition of each term.

2. Pages 7 to 10: Slope of land along the flow trajectory is one of important factors for the mechanism of the landslide and debris flow. The more detailed information of the slope is needed. Therefore, the cross sectional profile of the land along the flow is helpful for the discussion. Using the figure of the cross sectional profile, the authors should discuss the landslide and debris flow event.

3. Pages 10 and 19 The budget of sediment and water during this event is important for understanding the process of the landslide and debris flow. A schematic figure of this budget is helpful for the discussion.

4. Line 217, Page 10 Sernageomin (2018 c) found flow velocity 20 m/s at the Burritos River canyon. The authors should explain the method of estimating the velocity.

5. Line 331, pages 15 There are various factors in the basic equations describing the numerical simulation of the event, such as drag coefficient, basal friction angle, environmental resistant coefficient and fluid friction coefficient. Because the definition of these factors is not clear, it should be described in this paper.

6. 'pressures', line 342, page 16 and line 357, page 17 Generally speaking, the word 'pressure' is not used for open-channel flow but for pipe flow. Therefore, this word is incorrect.

7. 'a mudflow with a volume of water of 30%', line 346, Page 17 Does this mean the mudflow with sediment concentration of 70 %.?

---

## Referee Comment (RC3) · Anonymous Referee #3 · 30 Apr 2020

General Comments

The authors analyze a complex and composite mass movement event (a rockslide followed by a mudflow) occurred in Chile in 2017. They carry out the numerical simulation of the mud flow involved in the process, without addressing the issue of what actually triggered the rockslide event, which is not part of this research work. The authors compare the results obtained with the simulations carried out using two different available and already known software (FLO-2D and r.avaflow) that are used to back-calculate the propagation of the event and its main parameters. From this numerical study the authors derive a series of information, that they integrate/interpret with the results of

some field surveys and field tests, to provide an explanation of what has occurred. In particular, the study has helped to understand if the event was possible without the presence of a lake or a large water reservoir at the glacier that provided the amount of water needed to mobilize the mass. The results show that in fact the water available in the saturated soil.around the Burritos River was sufficient to transform the detritus flow into a mudflow.

The paper appears more as a technical paper describing and analyzing a case study than a research paper, and it should be presented as such, starting from the title. A possible suggestion would be for instance:

The mudflow disaster at Villa Santa Lucía in Chilean Patagonia: understandings and insights derived from numerical simulation and post event field surveys

The topic and the contents of the paper are certainly of interest for the scientific community and deserve publication, but the paper should be shortened and should focus on its real core. . Unfortunately, the paper is also written in an awkward English that does not help its understanding and clean reading. So the text requires substantial revision, possibly by a native speaker. I recommend a major revision, to be carried out also on the basis of the comments below.

Specific comments

Abstract - I would suggest to shorten the abstract and focus it on the main content of the paper, which is the interpretation of the catastrophic event and its causes based on field survey and numerical simulation. The reader expects to rapidly find in the abstract information regarding the main content of the paper, more than general comments on the treated issues. I have also reported some possible corrections to the English language, which are not intended, however, to be exhaustive because the entire paper requires substantial revision, possibly by a native speaker. Introduction -The same shortening suggested for the abstract should be done with the introduction, that should expand the focus regarding the interpretation of the mudflow event and its causes through field

surveys and simulation. For this purpose I would move lines 94-104 to the beginning of the introduction and then proceed with the other comments, substantially reduced. Methodology - The chapter should be restructured because it would be much better to have the content of the chapters 4.1 (Geotechnical results) and 4.2 (Soil Classification) presented all together in the chapter 3.1 (Fieldwork and Geotechnical sampling). This for two reasons: 1) the reader may have an idea of all the available geotechnical data collected in the field, finding them in the same place, without having to skip here and there in the paper 2) the reader would expect to find, within a chapter titled "results", the output of the calculations of the mathematical modelling, not the data deriving from field surveys and tests which concern more data acquisition than results of analysis or calculations. The titles of the chapters (or sub-chapters) should be restructured too: there are three chapters titled the same way, that is "numerical modeling": 3.2 Numerical Modeling 3.2 Numerical Modeling 5.2 Numerical Modeling This is somewhat misleading and does not reflect an describe the real content of each of these sections.

Conclusions. This is the best written part of the entire paper. It is simple, clear, straightforward. It declares what has been done, without any general digression. The entire paper should be restructured to adhere and to reflect what the authors write in their final conclusions, which should appear as the final synthesis of what has been written and developed before.

Technical corrections

See attached file.

Please also note the supplement to this comment:
https://www.nat-hazards-earth-syst-sci-discuss.net/nhess-2019-419/nhess-2019-419-RC3-supplement.pdf
* * *
[Figure]

**Supplement:**

[revised manuscript text omitted]

---

## Author Comment (AC1) · 21 Jun 2020

Dear Editor Dr. Daniele Giordan and Dr. Haruyuki Hashimoto

Thank you for your comments, we addressed them in our resubmitted version of this paper. In this document, we put the point-by-point responses to your comments.

With all the best

Dr. Marcelo Somos-Valenzuela

**Comment R2 (CR2) Dr. Haruyuki Hashimoto**

**CR2_1:** Pages 2 to 20: There are various technical terms expressing 'sediment-related disaster', such as rockslide, landslide, avalanche, hyper-concentrated flow, hyperconcentrated sediment flow, mudflow, debris flow, debris and mud flow, detritus flow, avalanche flow, mass flow, two-phase flow, non-Newtonian flow, and water-sediment mixture. These similar technical terms make us confused. In order to avoid the confusion, the authors should unify these similar words and then describe the definition of each term.

**Response to CR2_1:**
Thank you for this comment. We appoligize for being sloopy in the used of the terminology and finally used "landslide" to refer to the collapse of the wall that started the event. And mudflow when we refer to the event when water was added.
For the other terms,

- Hyper-concentrated sediment flows, mudflows and non-Newtonian flow are terms used in the Flo2D description Therefore, we provided in the text the references to consult the meaning (O'Brien and Zhao, 2004).
- For debris flow and avalanche flow, mass flow, two-phase flow, and water-sediment mixture in the description of R.Avaflow, we included the refence (Mergili, M., Pudasaini, 2019)
- Mud flow, Detritus flow Water-sediment mixture are not longer part of the document

References:

FLO-2D: Reference Manual, Nutrioso, AZ., 2018.

Mergili, M., Pudasaini, S. P.: r.avaflow - The mass flow simulation tool. r.avaflow 2.0 Software 2014-2019, http://r.avaflow.org/software.php, 2019.

**CR2_2:** Pages 7 to 10: Slope of land along the flow trajectory is one of important factors for the mechanism of the landslide and debris flow. The more detailed information of the slope is needed. Therefore, the cross sectional profile of the land along the flow is helpful for the discussion. Using the figure of the cross sectional profile, the authors should discuss the landslide and debris flow event.

**Response to CR2_2:**
We added a slope profile in Figure 1, which now looks as follow

[Figure]

Figure 1: Top left: Study area and extension of the inundation (South America and Los Lagos Region layer from https://tapiquen-sig.jimdo.com,.Top right: Burritos River blue line layer from http://datos.cedeus.cl/, background © ESRI). Bottom: Elevation profile and geological formation along the mudflow path.

**CR2_3:** Pages 10 and 19 The budget of sediment and water during this event is important for understanding the process of the landslide and debris flow. A schematic figure of this budget is helpful for the discussion.

**Response to CR2_3:** We appreciate your observation. After reviewing in detail our manuscript, we noted that figure 13 needs to be explained correctly. Now, we inserted the following sentence in the results section of r.avaflow

"We varied the percentage of water between 20% and 70%. The error for the heights, speeds calculated in each model are in Figure 9. Therefore, we propose that a mudflow with a 30% water volume could reproduce best the VSL event (Figure 10)."

We renumber Figure 13 as Figure 9, according to the figure prioritization suggested by reviewers.

**CR2_4:** Line 217, Page 10 Sernageomin (2018 c) found flow velocity 20 m/s at the Burritos River canyon. The authors should explain the method of estimating the velocity.

**Response to CR2_4:**

Sernageomin (2018) used Equation 1 from Johnson (1970) that empirically estimated the flow's velocity in a curve.

$$V = \sqrt{\left( g * R * \cos \alpha * \frac{\Delta h}{\Delta x} \right)}$$

**Equation 1**

Where:

      V= mean velocity  (m/s)
      g=gravity (m/s2)
      R=curve radius (m)
      $\alpha$=channel slope (°)

      The curves used for the calculationa re shown in the figure below:

[Figure]

Figure: Curves used by Sernageomin (2018) to estimate the velocity of the flow in that section of the Burritos River.

Table: Summary of the result from Sernageomin (2018)

| Curve | | C1 | C2 | C3 | |
|---|---|---|---|---|---|
| R | | 460 | 210 | 225 | |
| $\propto$ | | 2.121 | 1.626 | 10 | |
| Camber | $\Delta x$ | 145 | 55 | 57.4 | |
| | $\Delta h$ | 12 | 11 | 11 | |
| $\dfrac{\Delta h}{\Delta x}$ | | 0,08275862 | 0,2 | 0,19163763 | |
| $cos \propto$ | | 0,9993149 | 0,99959734 | 0,98480775 | |
| $V^2$ | | 372,820266 | 411,434266 | 416,141325 | |
| V (m/s) | | 19,3085542 | 20,2838425 | 20,3995423 | Average |
| V (Km/hr.) | | 69,5107952 | 73,0218329 | 73,4383522 | 71,9903268 |

**References:**

Sernageomin: Origen y efectos de la remoción en masa del 16.12.2017 que afectó la localidad de Villa Santa Lucía, comuna de Chaitén, Región de los Lagos, , 60, 2018.

Sernagoemin (2018) used: Johnson, A. M. (1970). Physical Processes in Geology: a Method for Interpretation of Natural Phenomena—Intrusions in Igneous Rocks, Fractures and Folds, Flow of Debris and Ice, Freeman, Cooper, and Co., San Francisco, California, 577 .

Since we did not do these calculations, we consider that we should not include them in our document and just cited the reference unless the editor thinks we should do so.

**CR2_5:** Line 331, pages 15 There are various factors in the basic equations describing the numerical simulation of the event, such as drag coefficient, basal friction angle, environmental resistant coefficient and fluid friction coefficient. Because the definition of these factors is not clear, it should be described in this paper.

**Response to CR2_5:**

We corrected the terms in the document that were poorly translated (see Response to SCR1_17) and we added the reference from r.avaflow. Therefore the descripton of the section where those terms were indicated reads as follow:

"To simplify the calibration, we divided the process into two. First, we set the sediment concentration by volume of the mudflow in 50% and change the  entrainment coefficient, basal friction angle, ambient drag coefficient and fluid friction coefficient. For the description of the parameters in r.avaflow see Mergili et al. (2017)."

Mergili, M., Fischer, J. T., Krenn, J. and Pudasaini, S. P.: R.avaflow v1, an advanced open-source computational framework for the propagation and interaction of two-phase mass flows, Geosci. Model Dev., 10(2), 553–569, doi:10.5194/gmd-10-553-2017, 2017.

**CR2_6:** 'pressures', line 342, page 16 and line 357, page 17 Generally speaking, the word 'pressure' is not used for open-channel flow but for pipe flow. Therefore, this word is incorrect.
**Response to CR2_6:**
One of the outputs from r.avaflow is flow pressure (see Table 1 from Mergili, M., Fischer, J. T., Krenn, J. and Pudasaini, S. P.: R.avaflow v1, an advanced open-source computational framework for the propagation and interaction of two-phase mass flows, Geosci. Model Dev., 10(2), 553–569, doi:10.5194/gmd-10-553-2017, 2017.)
However, we did not use pressure for the validation of the model and incorrectly used it here. So we deleted the word "pressure" from line 342 and 357.

**CR2_7:** 'a mudflow with a volume of water of 30%', line 346, Page 17 Does this mean the mudflow with sediment concentration of 70 %.?
**Response to CR2_7:**
Yes, it means that the sediment concentration is 70%

---

## Author Comment (AC2) · 21 Jun 2020

Dear Editor Dr. Daniele Giordan and Dr. Martin Mergili

Thank you for your comments, we addressed them in our resubmitted version of this paper. In this document, we put the point-by-point responses to your comments.

With all the best,

Dr. Marcelo Somos-Valenzuela

**Comments R1: Dr. Martin Mergili**

**General Comments R1 (GCR1):**

**GCR1_1:** Even though the paper is mostly written in an understandable way, there are various language issues which require substantial polishing (if possible, by a native speaker).
**Response to GCR1_1:**
We agree with your comment and had sent the paper for review to a professional Spanish-English translator. In the revised version you can see several corrections from Dr. Helen Lowry a native English speaker that provides this service.

**GCR1_2:** Title: the title sounds nice, but does it really describe the main content of the paper? I have the feeling that the paper is not so much about the conditions that potentially enabled the mud flow disaster, but rather on the propagation of the mud flow.
**Response to GCR1_2:**
We change the tittle to "The mudflow disaster at Villa Santa Lucía in Chilean Patagonia: understandings and insights derived from numerical simulation and post event field surveys" as it was suggested by the third reviewer (see **Response to GCR3_1**)

**GCR1_3:** Abstract: in my opinion, it could be condensed and a little bit more focused, but this is an issue of preference. The general statements at the beginning are maybe too long, given that the paper mainly describes a case study.
**Response to GCR1_3:**
We agree with the reviewer and modified the first paragraph from:
"The evaluation of potential mass wasting in mountain areas is a very complex process because there is not enough information to quantify the probability and magnitude of these events. Identifying the whole chain of events is not a straightforward task, and the impacts of mass wasting processes depend on the conditions downstream of the origin. Additionally, climate change is playing an essential role in the occurrence and distribution. Mean temperatures are continuously rising to produce long term instabilities, particularly on steep slopes. Extreme precipitations events are more recurrent as well as heat waves that can melt snow and glaciers, increasing the water available to unstabilized slopes"
To this:
"The evaluation of potential mass wasting in mountain areas is a very complex process because there is not enough information to quantify the probability and magnitude of these events. Identifying the whole chain of events is not a straightforward task, and the impacts of mass wasting processes depend on the conditions downstream of the origin."

**GCR1_4:** Introduction: the same as for the abstract. It starts with "Climate change", and the first tree paragraphs deal with change-landslide relations. Even though this is certainly an important topic, it is not the subject of the paper when looking at the methods, results, discussion and conclusions. If the authors would like to stay with the focus on climate change, this aspect has to be more strongly included in the main content of the paper. Otherwise, the introduction should be restructured and reformulated, shortening the part on climate change and coming more quickly to the core topic of the paper.

**Response to GCR1_4:**

We agree with the reviewer and took out all the paragraphs that may misleads the focus of the paper, which is not a climate change study and we rearrange the introduction.

Now the introduction reads:

"Introduction

Landslides processes are particularly dangerous in areas close to human settlements. They can affect nearby villages, directly destroying houses and taking human lives (Gariano and Guzzetti, 2016) or indirectly affecting the connectivity of remote areas (Winter et al., 2016). The impacts of landslides are a function of the size of the event but also of the conditions downstream. For example, glaciar lakes susceptible to overflow, as well as unstable valleys that, given the right soil matrix and water content, can mobilize and produce mudflows (Carey et al., 2011; Haeberli et al., 2013). Areas where glaciers are receding worsen this situation because they expose unstable hillslopes that can collapse as well as potentially create glacier lakes. Currently, baseline information availability still critical in austral zones of South America, especially in Northern Patagonia, with a low population density that has not encouraged rigorous evaluation. Moreover, in recent years landslides events have increased due to anthropic and climatic effects (Aldunce and González, 2009). Parallelly, northern Patagonia shows an increase in the population (INE, 2018) increasing the risk. Therefore, a better understanding of landslide dynamics like the chain of events type like mudflows is urgent.

In our contribution, we will evaluate the generation of a cascade of events associated with the Villa Santa Lucia mudflow in Northern Patagonia. In this study, we will evaluate the mechanisms that enable a landslide of $7x106$ m3 to evolve to the catastrophic mudflow that destroy Villa Santa Lucía in Chilean Patagonia, resulting in 22 people dead. The landslide, which may have been triggered by hydrometeorological conditions and destabilization of the wall around the receding Yelcho glacier, led to the generation of a hyper-concentrated flow at the head of the Burritos River that traveled around ten kilometers and affected 50% of the urban area of Villa Santa Lucia on December 16, 2017. The first observations indicated that the event was possible because of the presence of a glacier lake. However, field results do not allow to support this hypothesis in the area. Therefore, this study, which seeks to understand the conditions that enabled the event without the presence of a glacier lake, will have a two-fold application. First, it will allow us to understand the mechanisms of the chain of events leading to the 2017 mudflow in Villa Santa Lucia, and second, and probably most important, update the criteria for mapping risks associated with mudflows in Chilean Patagonia."

**Specific comments RC1 (SCR1)**

**SCR1_1:** L36: Please mention already here the region (southern Chile), many readers might not know the Yelcho mountain range.

**Response to SCR1_1:** We added (southern Chile) in the place indicated

**SCR1_2:** L47 (and in general): r-avaflow -> r.avaflow

**Response to SCR1_2:** We corrected the name of the software in several places.

**SCR1_3:** L49-50: You cannot determine the total water content from simulations and soil tests – you can just estimate it. Further, the precision given in the volume number (also in some other places) is too high, considering the uncertainties. In this case, 2.8 million m³ would be sufficient.
**Response to SCR1_3:** we changed from 2,789,500 to 2.8 million as suggested.

**SCR1_4:** L113: "… alluvial and river processes …": aren't alluvial processes also river processes?
**Response to SCR1_4:** Yes, they are. We deleted "alluvial" to avoid redundancy.

**SCR1_5:** L124 NO -> NW
**Response to SCR1_5:** Corrected

**SCR1_6:** Figure 2: Nice figure, but there are some hispanicisms is the legend … "Leyend" -> "legend"; "Hidrology" -> "Hydrology"; further, in the map itself the lake polygon should be deselected before exporting the map.
**Response to SCR1_6:** We fixed the figure

[Figure]

**SCR1_7:** L136: What would be the average annual rainfall in this area?
**Response to SCR1_7:** The average annual rainfall is
The original text read "According to the information provided by the "Dirección General de Aguas" (DGA), at the time of the event, the total annual rainfall was 3,650 mm, and in the 30 hours prior to the event the rainfall reached 124.8 mm, with a maximum intensity of 10.6 mm/hr at 16:00 hours on December 15, 2017."

Now it reads: "According to the information provided by the "General Water Directorate" (DGA in Spanish), the annual average rainfall in this area is 3,420 mm. At the time of the event, the total annual rainfall was 3,650 mm, and in the 30 hours prior to the event the rainfall reached 124.8 mm, with a maximum intensity of 10.6 mm/h at 16:00 hours on December 15, 2017."

**SCR1_8:** L156: A new section about the description of the event should start here.

**Response to SCR1_8:** We added a new section "Description of the event"

**SCR1_9:** L157: I would expect a little bit more information on the landslide-glacier interaction. Was there some glacier ice entrained, which was included in the flow downstream? This can be an important issue, even though it is not always straightforward to analyze its importance (see e.g.: *Mergili, M., Jaboyedoff, M., Pullarello, J., Pudasaini, S.P. (2020): Back-calculation of the 2017 Piz Cengalo-Bondo landslide cascade with r.avaflow. Natural Hazards and Earth System Sciences 20: 505-520. doi:10.5194/nhess-20-505-2020*)

**Response to SCR1_9:** Yes, this is an issue that we discuss, and as the reviewer mentioned, it is not a straightforward one. We did not quantified this, not have knowledge that somebody else did. However, we looked into satellite imagery to understand this interaction or to guess the importance of the ice entrainment. In the figure below, we put three images from the landslide area. The first image is an Aster image from 2010, the second is a Sentinel-2 image from 2017 before the event, and the third is a sentinel-2 image after the event.

When we compare Figures a and b, we can see that the glacier terminal shrunk about 250 meters from 2010 to 2017, exposing the walls in the north side. However, when we compare Figures b and c, we can see that the landslide moved away from the glacier terminal. The shape of the glacier terminal was not modified. The only option that we have left is that the end of the glacier is covered by debris. This is plausible; however, the volume remains unknown. Also, there is no indication of a glacier lake, as it was indicated in earlier studies. We can not ignore the fact that there was glacier entrainment, there is clear evidence in the field that there was. However, given the little information related to the volume of ice involved in this process, we seek to demonstrate using numerical modeling that the water available in the river banks and valley downstream was enough or nearly enough to generate a mudflow. We are aware though that there were other sources of water involved

We added extra information in the description of the event:
**"2.4    Description of the event**
On December 16, 2017, a 7 million cubic meter volcanic rock slide detached (Figure 3, point 1), falling next to the Yelcho glacier toe that sits on an intrusive formation that has a drop of about 80 meters. The glacier shows a retreat during the last decade. Sernageomin (2018c) indicated that there might have been a small lake and water available on top of the intrusive formation, but there is no conclusive information. Satellite images before the events do not indicate a lake above the intrusive unit. Also, there were indications that ice contributed to the mudflow. Remote sensing data do not show that the landslide felt on top of the glacier unless detritus had covered the glacier, although we can see ice on top of the flank that collapsed. However, we do not have data to prove or disregard the covered ice on the intrusive and how much water it contributed to the mix. Two million cubic meters of the material stayed above the intrusive, and five million cubic meters of material continued downstream, sliding on the intrusive unit at a slope of more than 70 degrees.  At this point, the mudflow reaches the Burrito River with high topographic differences. The flow continued along with the drainage network at high speed (Figure 1 and Figure 3, point 2), adding a significant amount of sediment into the mudflow.  The sediments are mainly associated with glaciolacustrine deposits (easy to mobilize) and ancient alluvials present in the valley and on the river walls."

[Figure]

Aster image, austral summer 2010

[Figure]

Sentinel S2 Austral summer before the event

[Figure]

Sentinel 2 after the event

**SCR1_10: L175:** Maybe add some brief information (1 sentence) about damages and casualties, readers might be interested in that. As the term "disaster" is included in the title of the paper, there should be at least some information on the socio-economic component.

**Response to SCR1_10**: After the dot we adeed "The mudflow destroyed of 50% of Villa Santa Lucía, killind 22 people and blocking two out of three access to the village Route 7 and Route 235".

**SCR1_11:** Figs. 4-8: They are very informative, but I recommend to put them together into one full-page figure with 7 panes or so.

**Response_to SCR1_11**: We put them all together in Figure 4

[Figure]

Figure 4: a) Area of slope failure slid and deposit; b) Channeled flow at the foothills; c and d) Aerial photo of non-channeled flood deposited in an old wetland captured with an InspireII UAV; e and f) Aerial photo of the channel in the last section before entering Villa Santa Lucia (left) captured with an InspireII UAV. Picture of the channel facing downstream (right); h and i) Villa Santa Lucia after the mudflow (From Sernageomin (2018c)).

**SCR1_12: L273:** "Topography". Further I recommend to shift the section 3.2.3 farther up, as it rather concerns data acquisition. In the place where it is now, it disturbs the flow of reading from the models to the parameterization.
**Response to SCR1_12:**
We move this section up and merge it with the first section of the metholodgy "3.1        Fieldwork, Geotechnical sampling and **topography**". So we pasted the paragraph and put it at the end of the 3.1 section.

**SCR1_13: Table 1:** you may round the inundation area and the flow velocity at Villa Santa Lucía – the numbers indicate a precision which is probably not justified by the data.
**Response to SCR1_13:**
We rounded it to 21 m/s

**SCR1_14: L289:** How did you perform the calibration? Did you just use an iterative optimization procedure ("trial and error"), or did you use some automated, systematic procedure? Please explain! These things are explained a little bit in Section 4 (results), but they should already be explained in the methods section.
**Response to SCR1_14:**
Yes, we did a trial and error calibration.
We added this in the calibration section in the methodology
It reads: "For the calibration of the models we used a **trial and error aproach seeking** too match the following three pieces of data available from Sernagomin: (1) Flood area, which was mapped after the event using aerial imagery (SAF, 2017); (2) Flow heights estimated by Sernageomin in Villa Santa Lucía and at the beginning of the canyon; and (3) Flow velocities in the canyon curve and at the beginning of Villa Santa Lucia (Figure 11). See Table 1 for the values."

**SCR1_15: 4.3.:** The heading "Numerical modelling" is misleading as, in this section, just the modelling domain and the calibration procedure are briefly described. This is something I would rather expect in the methods section, as it is not a result. Further: did you also consider simulating the entire event (including the initial landslide?) This could be an interesting task for the future and, as such, could be mentioned in the discussion. There is now the multi-phase model of Pudasaini (*Pudasaini, S.P., Mergili, M. (2019): A Multi-Phase Mass Flow Model. JGR Earth Surface. doi: 10.1029/2019JF005204*), which could also serve for the simulation of the interaction between the landslide and the glacier.
**Response to SCR1_15:**
We agree with the reviewer, we move this paragraph to Numerical modeling in the methodology section.
Thank you for the reference, this is a great resource for our future work. We considered to model the event from the initial landslide. However, we decided to simplify our simulations using the information that was generated by SERNAGEOMIN which is the national institution that deals with natural disaster associated to landslides. They quantify the amount of solid that continues downstream from the intrusive formation right below the glacier tongue. Also, we wanted to compare the different models we used. We had Flo2D and that model does not handle avalanches. But certainly this is a great study case that we will continue exploring.

**SCR1_16:** Section 4.4/Table 4: Again, some Hispanicisms (Si->Yes).
**Response to SCR1_16:** Corrected, our apologies for this type of mistakes.

**SCR1_17:** Further, there is no information about entrainment. Did you allow entrainment and, if yes, which value did you set for the entrainment coefficient? The "environmental resistance coefficient" is the "ambient drag coefficient", I think. Further, the Quasi Reynolds number and mobility number are 10^4.5 and 10^3. It is the logarithms which are given in the r.avaflow input.

**Response to SCR1_17**:

Yes, we did use entrainment and the entraiment coefficient is $10^{-5.75}$, the mobility number are indeed $10^{4.5}$ and $10^{3}$. Our mistake was that we translated entrainment to Spanish and then back to English as "drag". We corrected this in the document as well as the symbol and the base 10.

We modify Table 4, so now it look like this.

| | | | |
|---|---|---|---|
| Solid density $[g/cm^3]$ | 2.400 | Terminal Velocity | 1 |
| Liquid density $[g/cm^3]$ | 1.000 | Contribution parameter S-L drag resistance | 0.500 |
| Virtual mass | 0.500 | Fluid friction coefficient | 0.002 |
| Hydrograph | No | Output writing time (s) | 10 |
| Diffusion control | Yes | Internal friction angle | 24 |
| Conservation of volume | Yes | Particle Reynolds number | 1 |
| Surface control | Yes | Exponent for drag | 1 |
| Viscous shear coefficient of the fluid | 0 | Quasi Reynolds Number | $10^{4.5}$ |
| Solids concentration distribution with depth | 0 | Mobility Number | $10^{3}$ |

**SCR1_18:** Fig. 12 (and some other places): flow high -> flow height

**Response to SCR1_18**: We replaced Figures 12-16, now they are Figures 8-12, see also **Response to SCR1_20.**

[Figure]

[Figure]

**SCR1_19:** L341/342: Better: "… We varied the percentage of water between 20% and 70% …" – the formulation as it is now is misleading.
**Response to SCR1_19:**
We modified this sentence, thank you for the suggestion.

**SCR1_20:** Fig. 14, legend: revise the thresholds: e.g., to which class would a flow height of 49.95 m belong?
**Response to SCR1_20:**

We modified Figure12 (now figure 8):

[Figure]

Figure 8: R.avaflow modeling results for a concentration by volume of 50% (Background © ESRI).

We modified Figure14 (now figure 10):

[Figure]

Figure 10: Best simulation results using the calibrated parameters for r.avaflow and 30% water content (Background © ESRI)

And we modified Figure 16 (now Figure 12)

[Figure]

Figure 12: Result for Glenwood 1 using a 40% concentration of water by volume (Background © ESRI)

**SCR1_20:** Section 5.2.: You should also briefly mention the limitations of your calibration due to issues of equifinality (e.g. *Beven, K. (1996). 12 Equifinality and uncertainty in geomorphological modelling. In The Scientific Nature of Geomorphology: Proceedings of the 27th Binghamton Symposium in*

*Geomorphology, Held 27–29 September 1996 (Vol. 27). John Wiley & Sons.*), and the multi-dimensional parameter space (e.g. *Saltelli, A., & Annoni, P. (2010). How to avoid a perfunctory sensitivity analysis. Environmental Modelling & Software, 25(12), 1508-1517.*).

**Response to SCR1_10**:
Thank you for these excellent references that we did not have considered. They certainly open a new way of thinking on how to carry out calibration for these very complex processes.
So we added this paragraph in section 5.2

"For the calibration, we calculated the standard deviation from the modeling results and the parameters from Table 1.  The parameter combination that results in less standard deviation is considered the best parameterization for the particular software used. Our procedure, as described in Saltelli and Annoni (2010), corresponds to a one-factor-at-a-time type of calibration where we change one parameter at a time manually, trying to match Table 1. The limitations of this procedure, according to Saltelli and Annoni (2010), are: Its efficiency is poor. The method does not capture the interaction of the factors because that would require the movement of more than one element at a time; therefore, the approach assumed that all the variables are independent, and the processes are linear, which is not usually the case. Another limitation of our calibration is the potential presence of equifinality. We did not test if different combinations of parameters would provide similar performance. We selected one best combination of parameters without checking the degrees of freedom that these variables may have had to replicate the observations (Beven, 1996)."

*SCR1_21: L400: The water content leading to the empirically most adequate results was approx. 30%. Is this fraction also plausible from a physical point of view, and from the observations? Please briefly elaborate on this aspect more explicitly (some indirect information is given in the paragraph below).*
**Response to SCR1_22:**
It is possible from the physical point of view and from our observations. We added in the section Source of water "Our analyses determined that in the area scoured, the soil has a porosity of  84.18%. Similar values were reported by Cuevas et al. (2013) in volcanic soils in the south of Chile. From our observations, we also notice the ground is saturated all the time. We did fieldwork after a couple of weeks of no rain, and the whole area around the river was saturated; indeed, our soil sample had a saturation of 97.4%.  Therefore, using the results from r.avaflow, we estimated that a volume of 3.402.100 $m^3$ was scoured. Consequently, there was the potential of adding roughly 2.8 million $m^3$ of water and 600,000 $m^3$ of soils to the avalanche, which evolved to a mudflow due to the water added to the event."

Cuevas, J., Horn, R., Seguel, O. and Dörner, J.: Hydraulic conductivity variation in chilean volcanic soils due to wheeling and management, J. Soil Sci. Plant Nutr., 13(3), 756–766, doi:10.4067/S0718-95162013005000060, 2013.

*SCR1_23: L404, 405: Oh, you computed entrainment! This is good, but it is mentioned here for the first time (unless I overlooked it). You should appropriately address this important aspect also in the methods and the results sections.*
**Response to SCR123:**
*We apologize for the poor translation. You did not overlooked, the words use in our manuscrit were incorrect. In the methodology, it was mentioned that we considered the "drag of material" (line 256) now it reads "The entraiment of material along the flow path was also considered." Then in line 262, we* change "dragged heights" for "scoured heights" which refers to the basechanges file from r.avaflow. *Therefore in section 4.1 we mention this in the first paragraph*

And we include this topic in the discussion, see **Response to SCR1_22**

"4.1     R.Avaflow
To simplify the calibration, we divided the process into two. First, we set the sediment concentration by volume of the mudflow in 50% and change the  entrainment coefficient, basal friction angle, ambient drag coefficient and fluid friction coefficient. For the description of the parameters in r.avaflow see Mergili et al. (2017). Table 4 shows the first set of parameters used."

In the result section we reporte the calibrated value for the entrainment coefficient as $10^{-5.75}$

*SCR1_24: L409: Now, the entrainment of glacier ice comes into play! You have to introduce this aspect already in the event description (see comment above).*
***Response to SCR1_24:***
*We modified the description of the event. See* **Response to SCR1_9** *where we addressed this issue.*

*SCR1_25: L420: FLO-2D is not a freely available software.*
***Response to SCR1_25:***
*We were refereeing just to r.avaflow, so we added sentence to make this point clear "*The combination of geotechnical tests and R.avaflow, which is a freely available computational software,…"

*SCR1_26: Conclusions, second paragraph: I would rather suggest to move this text to the discussion. The conclusions should rather focus on the key messages from your work (just extend what is written in the first paragraph).*
***Response to SCR1_26****:*
We moved part of the second paragraph of the conclusion to the end of the discussion. The other part, we move it to the beginning of the conclusion section which now reads as follow:

**"6  Conclusions**
Given the complexity and the potential increase in the future of extreme event occurrence that can trigger landslides, we suggest that hazard studies should consider the structural conditions present in the area of influence of the mudflows. Soil characteristics ought to be included because they may be a crucial factor amplifying the impacts of local events triggered by hydroclimatic events. Such was the case of Villa Santa Lucia, where the water necessary to fluidize the mudflow mixture was in the soil in a volatile system that was easy to mobilize.
The combination of geotechnical tests and r.avaflow, which is a freely available computational software, to model mudflow are useful tools to characterize and reproduce mass wasting events, such as the mudflow occurred on December 2017 in Villa Santa Lucia in Chile. Our results present the possibility of open-source software implementation to represent mudflow events in the Patagonian Andes with a good performance. These types of studies will allow for the integration of better methodologies to enhance risk scenarios related to mudflow events in active subduction zones like the Andes."

---

## Author Comment (AC3) · 21 Jun 2020

Dear Editor Dr. Daniele Giordan and anonymous reviewer

Thank you for your comments, we addressed them in our resubmitted version of this paper. In this document, we put the point-by-point responses to your comments.

With all the best

Dr. Marcelo Somos-Valenzuela

**Comments R3: Anonymous**

**General Comments R3 (GCR3):**
**GCR3_1:**
The paper appears more as a technical paper describing and analyzing a case study than a research paper, and it should be presented as such, starting from the title. A possible suggestion would be for instance: The mudflow disaster at Villa Santa Lucía in Chilean Patagonia: understandings and insights derived from numerical simulation and post event field surveys
**Response to GCR3_1:**
We take you recommendation and now the title of the paper is "The mudflow disaster at Villa Santa Lucía in Chilean Patagonia: understandings and insights derived from numerical simulation and post event field surveys"

**GCR3_2:**
The topic and the contents of the paper are certainly of interest for the scientific community and deserve publication, but the paper should be shortened and should focus on its real core. Unfortunately, the paper is also written in an awkward English that does not help its understanding and clean reading. So the text requires substantial revision, possibly by a native speaker. I recommend a major revision, to be carried out also on the basis of the comments below.
**Response to GCR3_2:**
We appreciate that you considered that this works deserves to be published after the suggested correction are made. We sent this document for professional English translation and edition. Which you can check in the modified document. We also modified the summary and introduction following your suggestion and Reviewer 1's suggestions. Please see **Response to GCR1_3 and Response to GCR1_4**

**Specific comments R3 (SCR3)**

**SCR3_1:** Abstract - I would suggest to shorten the abstract and focus it on the main content of the paper, which is the interpretation of the catastrophic event and its causes based on field survey and numerical simulation. The reader expects to rapidly find in the abstract information regarding the main content of the paper, more than general comments on the treated issues. I have also reported some possible corrections to the English language, which are not intended, however, to be exhaustive because the entire paper requires substantial revision, possibly by a native speaker.
**Response to SCR3_1:**
Thank you for all the suggestion to the original document, we have included them and also sent the paper for English professional revision. We also shorthened the abstract please see **Response to GCR1_3 (response to general comment 3 from reviewer 1)**

**SCR3_2:** Introduction -The same shortening suggested for the abstract should be done with the introduction, that should expand the focus regarding the interpretation of the mudflow event and its causes through field surveys and simulation. For this purpose I would move lines 94-104 to the beginning of the introduction and then proceed with the other comments, substantially reduced.

**Response to SCR3_2**:
We added a short paragraph before the suggested place from Reviewer 3. We reduced the introduction from 947 to 345 words. Please see **Response to GCR1_4.**

**SCR3_3:** Methodology - The chapter should be restructured because it would be much better to have the content of the chapters 4.1 (Geotechnical results) and 4.2 (Soil Classification) presented all together in the chapter 3.1 (Fieldwork and Geotechnical sampling). This for two reasons: 1) the reader may have an idea of all the available geotechnical data collected in the field, finding them in the same place, without having to skip here and there in the paper 2) the reader would expect to find, within a chapter titled "results", the output of the calculations of the mathematical modelling, not the data deriving from field surveys and tests which concern more data acquisition than results of analysis or calculations.

**Response to SCR3_3**:
We partially agree with this coments. Chapter 4.1 and 4.2 are part of the results of our work so we think that for that reason they belong to the result section. However, it is true that the paper looked disorganized since it provided bits of the same information in different sections which gets confusing. So we accept the suggestion and moved the sections 4.1 and 4.2 from the results to the methodology limiting the results section to the results from the modeling work

*SCR3_4: The titles of the chapters (or sub-chapters) should be restructured too: there are three chapters titled the same way, that is "numerical modeling": 3.2 Numerical Modeling 3.2 Numerical Modeling 5.2 Numerical Modeling This is somewhat misleading and does not reflect an describe the real content of each of these sections.*

*Response to SCR3_4:*
Following the comment above (**SCR3_3**), we reduced the results section and eliminate the secion 4.2 Numerical modeling. We also renamed the section 5.2 to "Back-calculation of the mudflow"

**SCR3_5:** Conclusions. This is the best written part of the entire paper. It is simple, clear, straightforward. It declares what has been done, without any general digression. The entire paper should be restructured to adhere and to reflect what the authors write in their final conclusions, which should appear as the final synthesis of what has been written and developed before.

**Response to SCR3_5**:
We hope that after the modifications the entire document reads as it does the former conclusion section.

**Extra comments:**
We also included all the English suggestions from the nhess-2019-419-RC3-supplement into the revised document that is included in a separate file.

---

## Referee Report (RR1)

[referee-annotated manuscript omitted]

---

## Author Response (AR2)

July 23, 2020

Dear Editor Dr. Daniele Giordan

Thank you for accepting our paper with minor correction. In this iteration, we submit our reviewed document with the suggestions from reviewer 3. We also sent the text to review from a native English speaker. We also include the figures a version with the changes, and a version with the changes accepted in pdf format.

With all the best,

Dr. Marcelo Somos-Valenzuela

[revised manuscript text omitted]